# In Vivo Predictive Dissolution (IPD) for Carbamazepine Formulations: Additional Evidence Regarding a Biopredictive Dissolution Medium

**DOI:** 10.3390/pharmaceutics12060558

**Published:** 2020-06-17

**Authors:** Marival Bermejo, Jessica Meulman, Marcelo Gomes Davanço, Patricia de Oliveira Carvalho, Isabel Gonzalez-Alvarez, Daniel Rossi Campos

**Affiliations:** 1Department of Engineering, Pharmacy and Pharmaceutical Technology Area, Facultad de Farmacia, University Miguel Hernandez de Elche, San Juan de Alicante, 03550 Alicante, Spain; mbermejo@umh.es; 2Faculty of Pharmaceutical Sciences, University of Campinas—UNICAMP, Campinas, 13083-871 São Paulo, Brazil; jessicameulman@gmail.com; 3Postgraduate Program in Health Sciences, Universidade São Francisco, Bragança Paulista, 12916-900 São Paulo, Brazil; davanco.marcelo@gmail.com (M.G.D.); patricia.carvalho@usf.edu.br (P.d.O.C.); camposrossi@gmail.com (D.R.C.)

**Keywords:** carbamazepine, in vitro in vivo correlation, dissolution, biopredictive, bioequivalence, biowaiver

## Abstract

The aim of the present study was to bring additional evidence regarding a biopredictive dissolution medium containing 1% sodium lauryl sulphate (SLS) to predict the in vivo behavior of carbamazepine (CBZ) products. Twelve healthy volunteers took one immediate release (IR) dose of either test and reference formulations in a bioequivalence study (BE). Dissolution profiles were carried-out using the medium. Level A in vitro–in vivo correlations (IVIVC) were established using both one-step and two-step approaches as well as exploring the time-scaling approach to account for the differences in dissolution rate in vitro versus in vivo. A detailed step by step calculation was provided to clearly illustrate all the procedures. The results show additional evidence that the medium containing 1% SLS can be classified as a universal biopredictive dissolution tool, and that both of the approaches used to develop the IVIVC (one and two-steps) provide good in vivo predictability. Therefore, this biopredictive medium could be a highly relevant tool in Latin-American countries to ensure and check the quality of their CBZ marketed products for which BE studies were not requested by their regulatory health authorities.

## 1. Introduction

Carbamazepine (CBZ) is a drug widely used in the treatment of epilepsy and trigeminal neuralgia. CBZ is a Biopharmaceutics Classification System (BCS) class II drug with a low aqueous solubility and high permeability [1]. Therefore, the drug is poorly soluble in aqueous media [2]. Besides its poor aqueous solubility, other attributes such as a narrow therapeutic index and relatively high variability have been recognized as obstacles for CBZ product development for bioequivalence proposals [3].

In vitro–in vivo correlations (IVIVC) are widely used tools in biopharmaceutic research in order to speed up the product development for quantifying the in vivo release, evaluating formulation-related effects on absorption and as a tool for setting in vitro dissolution specifications [4,5]. The FDA recognizes four levels of IVIVC: Level A, B, C, and multiple Level C. The most desired is the Level A category of IVIVC. It is defined as a point-to-point relationship between in vitro dissolution and the in vivo response, such as plasma drug concentration or amount of drug absorbed [4]. Level A regression may be complicated due to differences in time scales as well as the lack of coincident times of similar release in vitro and in vivo [6,7]. Discrepancies between in vivo and in vitro times are observed by faster in vitro dissolution compared to in vivo release or by differences in shape between the two curves. In both cases, a direct relationship between in vitro and vivo data cannot be set up simply. The most used approach to determine time scaling is the so-called Levy plot. Times at which in vivo and in vitro the same percentage is absorbed and dissolved, respectively, are plotted in the Levy plot [7,8].

However, the major objective of a validated IVIVC is to use in vitro dissolution data to predict in vivo performance, serving as a surrogate for an in vivo bioequivalence (BE) study, e.g., supporting a biowaiver approach. In classical two-stage approaches, fraction dissolved, obtained from in vitro dissolution profiles is typically used together with corresponding in vivo fraction absorbed obtained by deconvolution of observed plasma concentrations [6]. On the other hand, the one-stage approach uses the in vitro dissolution data and pharmacokinetic characteristics of the drug to obtain the adequate link function of the plasma drug concentrations, directly by convolution [9].

Some authors have already described a universal (for CBZ) biopredictive feature of dissolution medium containing 1% sodium lauryl sulphate (SLS), which is available in the USP Pharmacopeia [10,11]. This medium has been used for quality control proposal and additionally as a biorelevant dissolution medium since Kovaĉević, I. et al. [1] performed gastrointestinal simulations as well as established IVIVC for both immediate and modified release formulations of CBZ. Additionally, González-García, I. et al. [12] recently applied the IVIVC approach using the USP medium described above, showing the biopredictive feature of this medium, which causes the immediate release of formulations with conventional excipients, even if different batches and in vivo/in vitro studies are combined. The biopredictive dissolution method is defined by Suarez-Sharp et al. [13] as a set of testing conditions in which in vitro dissolution profiles can predict the pharmacokinetic profiles.

The aim of the present study was to perform an IVIVC for two CBZ immediate release formulations (used in a pilot BE study) using both a one-step and two-steps approaches as well as to explore the time-scaling approach to account for the differences in dissolution rate in vitro versus in vivo. A detailed step by step calculation was provided to clearly illustrate all the procedures. Moreover, this paper intends to bring additional evidence of using of IVIVC-based biowaiver for BCS class II drugs.

## 2. Materials and Methods

### 2.1. Formulations 

CBZ 400-mg tablets (test formulation) and Tegretol^®^ (reference formulation) were purchased in the local Brazilian market. 

### 2.2. Bioanalytical Method 

Plasma concentrations of CBZ were determined using liquid chromatography with tandem mass spectrometry assay—LC-MS/MS (Waters) with electrospray ionization source in positive mode. Carbamazepine-d8 was used as internal standard. The chromatographic separation was performed at 40°C using a column X-Bridge C18 (4.6 × 50 mm, 3.5 μm) and flow rate of 0.50 mL/min. The quantification was performed by using multiple reaction monitoring (MRM) mode of the transitions at m/z 237.051 > 194.240 and m/z 244.600 > 202.300 for CBZ and carbamazepine-d8, respectively. The analytes were extracted from plasma using protein precipitation with methanol as solvent. The mobile phase used was 0.1% formic acid and methanol at a 35:65 ratio (v/v) and the injection volume was 5 μL and the total run time set as 4 min. 

The bioanalytical method was validated in compliance with ANVISA guidance for bioanalytical method validation [14] and FDA Bioanalytical Method Validation Guidance for Industry [15].

### 2.3. Bioequivalence Study 

The bioequivalence study was approved by the Research Ethics Committee (protocol number 3.085.454). All procedures were conducted in the Clinical Trials Center in accordance with the principles of Good Clinical Practice guidelines [16], the Declaration of Helsinki [17], and Resolution 466/2012 (Ministério da Saúde, Brazil) [18,19]. Written informed consent from all participants was obtained prior to the enrollment.

All participants were aged between 18 and 50 years with a body mass index between 18.5 and 29.9 kg/m^2^. All subjects showed good health conditions or the absence of significant diseases after assessment of medical history, verification of vital signs, physical examination, electrocardiogram and routine laboratory tests. The study design was randomized, single dose, fasting, two-period, two-sequence crossover with a 14-days washout period. Twelve adult healthy subjects of both genders were enrolled in the study and eleven (6 women and 5 men) subjects completed the two study periods. The blood samples (7.5 mL each) were obtained at 0 h (pre-dose) and at 1.00, 2.00, 3.00, 3.50, 4.00, 4.50, 5.00, 5.50, 6.00, 8.00, 10.0, 12.0, 24.0, 48.0 and 72.0 h post-dose during each period. Each blood sample was collected in EDTA tubes as anticoagulant at each time point. Collected blood samples were centrifuged immediately (3500 rpm for 10 min at 4 °C) and plasma was separated and stored frozen at −20 °C with appropriate labeling until sample analysis.

### 2.4. Pharmacokinetic and Fraction Absorbed Analysis

The pharmacokinetic parameters were obtained from the curves of plasma concentration versus time for CBZ and statistically compared for determination of bioequivalence, using Phoenix WinNonlin software version 8.0 (Bioequivalence Wizard module). The area under the curve from zero to the last quantifiable concentration (AUCt) was calculated by the trapezoidal method, and the area under the curve from zero to infinity (AUC∞) was calculated by the formula AUCt+ (Cn/kel), where Cn was the last quantifiable plasma concentration. Due to the long elimination half-life (t_1/2_) of CBZ, it was considered area under the curve (AUC) truncated (72 h). The elimination rate constant (kel) was determined by the elimination phase of the graph of log plasma concentration versus time. The t_1/2_ was defined using the equation t_1/2_ = Ln(2)/kel. The maximum plasma drug concentration (Cmax) was obtained directly from the experimental data, as well as the time of the occurrence of Cmax (tmax).

Bioequivalence assessment was based on predefined acceptance criteria of 80.00–125.00% for the 90% confidence interval for the ratio of the test and reference products for the log-transformed data of AUC and Cmax, as recommended by (19), and FDA [20]. An ANOVA was performed for the primary parameters estimated (Cmax and AUC) to evaluate formulation, sequence, and period as fixed effects and to estimate the residual variance to construct the confidence intervals [21]. More detailed calculations are reported in Appendix A.

The drug absorption was estimated using numerical deconvolution method from Wagner–Nelson. This mass balance method, with First order elimination, was employed because it is the most suitable for one-compartment drugs and has been shown to adequately describe the pharmacokinetic absorption profile of CBZ:(1)$$Fabs=\frac{A_{t}}{A^{\infty }}=\frac{C_{t}+kel\times AUC_{t}}{kel\times AUC^{\infty }\;\;}$$

Equation (1) is the Wagner–Nelson equation that represents the fraction absorbed of the bioavailable dose at time *t*. *Fabs* is the fraction absorbed; *A_t_* is the drug amount absorbed at time *t*; *A^∞^* is the drug amount absorbed at infinite time, *C_t_* is the drug concentration at time *t*; *kel* is the elimination rate coefficient; *AUC_t_* is the area under the curve from time zero to the last quantifiable concentration and *AUC^∞^* is the area under the curve from zero to infinity.

### 2.5. In Vitro Dissolution Testing and Modelling

Dissolution study was performed in the PhEur/USP rotating paddle apparatus at 75 rpm using 900 mL of dissolution medium. The medium was a 1% sodium lauryl sulfate (SLS) aqueous solution. Dissolution study was performed with the same batches used in the in vivo study. CBZ concentrations on the dissolution samples were analyzed by HPLC. Samples of 5mL were taken at 5, 10, 15, 30, 60 and 120 min. The experiment was performed with twelve tablets for each formulation. 

First order and Weibull models were fitted to the data (fractions dissolved, *F_diss_*) of each formulation. 

Weibull equation
(2)$$F_{diss}=100\cdot \left(1-e^{\left(-\frac{\left(t^{\beta }\right)}{\alpha }\right)}\right)$$
where *F_diss_* are fractions dissolved and *β* and *α* are the Weibull parameters

First order equation
(3)$$F_{diss}=100\cdot \left(1-e^{\left(-kd\cdot t\right)}\right)$$
where *kd* is the dissolution rate constant. First order equation is a particular case of Weibull model when *β* = 1; then *kd* = 1/*α*

The best model was selected based on the correlation coefficient of experimental versus predicted values, the Akaike’s information criteria (AIC), and the residual variance comparison with Snedecor’s F tests [22,23].

Fitting procedures were performed in Excel with DDsolver add-in [24].

With the Weibull parameters of each profile, it is possible to calculate the time needed for the dissolution of any desired fraction dissolved with Equation (4):(4)$$t_{vitro}=\sqrt[{\beta }]{\alpha *\left(-1\right)*ln\frac{100-F_{abs}}{100}}$$
where *β* and *α* are the Weibull parameters (from Equation (2)) and *F_abs_* the corresponding fraction absorbed (dissolved) in vivo.

### 2.6. IVIVC Two-Step Aproach

In the two-step approach fractions dissolved and absorbed at the same time points are correlated.

The scheme of the calculations for the two-step approach is represented in Figure 1.

As the time scale was different on the in vitro and the in vivo assays, a Levy plot was necessary. To construct the Levy plot, a dissolution model is fitted to fraction dissolved data (step 1). Fraction-absorbed values were interpolated into the dissolution model and the equivalent in vitro times were obtained with Equation (4) and the Weibull parameters from each dissolution profile (step 2). In vitro and in vivo times were represented together and the Levy plot correlation parameters estimated (in the present paper, a linear one) (step 3). In vivo times were included up to 24 h for the test formulation and up to 12 h for the reference as later on the relationship was no longer univocal. 

With the Levy equation, the in vitro sampling times were converted to their equivalent in vivo times (step 4). The objective of this procedure is to have the dissolution profile and the absorption profile in the same time-scale to check if they are directly superimposable. 

As the dissolution in vitro profiles are scaled with the equivalent in vivo time, the Weibull model was fitted again to the data to obtain the scaled Weibull parameters (step 5).

The new Weibull parameters were used to estimate the fraction dissolved at the original in vivo times (step 7) so that, finally, fractions absorbed and dissolved at the same times could be plotted and the IVIVC linear relationship characterized (step 8).

To determine the predictability of the IVIVC correlation, predicted fractions absorbed (y) from the linear IVIVC (using the fractions dissolved (*x*)) were back-transformed into plasma concentrations using the following equation from Gohel et al. [25].
(5)$$C_{t+1}=\frac{\left(\frac{2\cdot \Delta F_{abs}\cdot D}{V_{d}}\right)+Ct\cdot \left(2-Kel\cdot \Delta t\right)\;\;}{\left(2+Kel\cdot \Delta t\right)}$$
where *C_t+1_* is the plasma concentration at time (*t* + 1) and then *C_t_* is the plasma concentration in the previous sampling time, t. Δ*t* is the time interval between a sampling time and the next one and *F_abs_* are the predicted fractions absorbed from the IVIVC correlation. *D* is the CBZ dose, *kel* the elimination rate constant and *Vd* the apparent distribution volume. This *Vd* values was estimated with the following equation.
(6)$$Vd=\frac{D}{\left(AUC_{0}^{\infty }\cdot kel\right)}$$

*kel* (0.0159 h^−1^) and *AUC* (area under the curve from time zero to infinity) (371,958.8 ng/mL∗h) values were the average values from both formulations. The estimated or apparent Vd (Distribution volume divided by the bioavailability F (Vd/F)) was 67,698 mL.

### 2.7. IVIVC One-Step Approach

In the one-step approach, the fractions dissolved from test and reference were directly convoluted with the adequate scale factors with the pharmacokinetic parameters of CBZ (kel and Vd) to estimate plasma levels. 

A system of differential equations was set up to simultaneously fit fractions dissolved (to a Weibull equation as described previously) and plasma levels.

Four differential equations were defined, two for each formulation, representing the fraction dissolved (dFdissx/dt) and the plasma levels (dCx/dt) where x is reference or test
(7)$$\frac{dFdissTest}{dt}=\left(betaTest\times Fdissmax\times \left(t^{betaTest-1}\right)\right)\times e^{\left(\frac{\left(-\left(t^{betaTest}\right)\right)}{alfaTest}\right)})/alfaTest$$
(8)$$\frac{dFdissRef}{dt}=\left(betaRef\times Fdissmax\times \left(t^{betaRef-1}\right)\right)\times e^{\left(\frac{\left(-\left(t^{betaRef}\right)\right)}{alfaRef}\right)})/alfaRef$$
where *Fdissmax* is the maximum dissolved fraction which was fixed to 1 (100%); *betaTest*, *alfaTest*, *betaRef* and *alfaRef* were the parameters of the Weibull equation for the in vitro dissolution for both Test and Reference formulations respectively. These two differential equations are simply the derivatives of Equation (2). 

In order to estimate the plasma levels, the differential equation describing how drug amounts in the body change with time (dMass/dt) is designed taking into account the input rate and the elimination rate.

In a one-compartment pharmacokinetic model with First order elimination, the elimination rate in terms of drug mass is kel*Mass. The input rate into the system is limited by the drug dissolution (that is drug cannot be absorbed until dissolved), thus dissolution rate (defined by the two previous equations) corresponds to the input rate into the system (body). There are two modifications needed to estimate the in vivo input rate. In Equations (7) and (8), the dissolution rate is defined in terms of fractions dissolved, then to estimate the in vivo mass dissolved the previous equations are multiplied by the dose to get the input rate.
(9)$$\frac{dMassTest}{dt}=\left(Dose\times betaTvivo\times \left(t^{betaTvivo-1}\right)\right)\times e^{\left(\frac{\left(-\left(t^{betaTvivo}\right)\right)}{alfaTvivo}\right)})/alfaTvivo-kel\times MassTest$$
(10)$$\frac{dMassRef}{dt}=\left(Dose\times betaRvivo\times \left(t^{betaRvivo-1}\right)\right)\times e^{\left(\frac{\left(-\left(t^{betaRvivo}\right)\right)}{alfaRvivo}\right)})/alfaRvivo-kel\times MassRef$$

The Weibull parameters for the in vivo dissolution of the Test formulation were *betaTvivo* and *alfaTvivo*. *betaRvivo* and *alfaRvivo* were the Weibull parameters for the in vivo dissolution of the Reference formulation. *MassTest* and *MassRef* correspond to the amounts in plasma of CBZ from Test and Reference formulation and *kel* was the elimination rate constant.

Finally, amounts in plasma (Mass) are transformed in plasma levels dividing the amounts by the distribution volume (Vd) previously defined.

The link between in vitro and in vivo dissolution was established through a scaling factor (as in the two-step approach) of the Weibull parameters. The scaling factors (*scalfa* and *scbeta*) must be the same for both formulations.
(11)alfaTvivo=scalfa×alfaTest
(12)alfaRvivo=scalfa×alfaRef
(13)betaTvivo=scbeta×betaTest
(14)betaRvivo=scbeta×betaRef

Fitting procedures were carried out with Phoenix WinNonlin (version 8.0) and Berkeley Madonna 9.1.19 with similar results. Codes in both software are provided in the Appendix A.

## 3. Results and Discussion

Figure 1 shows the average plasma levels for both assayed formulations (test and reference). Figure A1 in the Appendix A represents the individual plasma levels, and Figure A2 includes the average levels with error bars.

When the Tmax variability across individuals is very high (due to, for instance, a highly variable lag time (Tlag)), it could be possible that the average plasma profile does not represent the individual behavior, which complicates the development of an IVIVC. Nevertheless, if the Tlag and Tmax of all subjects and of the mean curves are close together (as it is the case in the present data), the use of a mean curve will not dramatically modify the results [6]. Consequently, it was decided that average plasma levels for deconvolution would be used.

As there is no intravenous CBZ data available, we could not identify the pharmacokinetic compartmental model of the drug. Nevertheless, in the literature, CBZ oral profiles have been successfully described with a one-compartment model with reasonable accuracy [26,27,28].

For that reason, the Wagner–Nelson deconvolution method was selected to estimate bioavailable fractions. 

Figure 2 represents the fractions absorbed (actually bioavailable fractions) obtained by the Wagner–Nelson method. The plot is restricted up to 40 h to clearly show the differences on the initial times.

In spite of the fact that the CBZ true compartmental pharmacokinetic model might be a two-compartment one, due to its lipophilicity ad long half-life, the Wagner–Nelson mass balance did not detect any relevant bias. As it can be observed in Figure 3, the fractions absorbed smoothly increase up to 100% without surpassing this value, as it is frequent when the procedure is applied to a two-compartment drug. This fact confirms the suitability of the one-compartment pharmacokinetic model to describe the CBZ absorption profile. 

### 3.1. Bioequivalence Results

Table 1 summarized the results of the pilot bioequivalence study (*N* = 11).

Statistical analysis of the bioequivalence study was performed with the adequate procedures for an average bioequivalence cross over design (two periods, two sequences, two formulations). The observed residual variability was low (less than 20%) thus the confidence interval calculation did not need any correction and the acceptance range was 0.8 to 1.25. 

As it could be expected for a class II drug (low solubility, high permeability), the observed failure is on the pharmacokinetic parameter associated with the rate of absorption, Cmax. In a low solubility drug, the dissolution process can be the limiting factor for absorption, consequently formulation factors (as excipients) and drug factors (as particle size) affecting dissolution rate can influence Cmax. As long as the dissolution of both formulations is completed during transit time, the extent of absorption, due to the high permeability, will be complete and similar (reflected in equivalent AUC values). In summary, Table 1 reflects the bioequivalence failure in absorption rate conditioned by the drug products dissimilar in vivo dissolution, leading to different input rates in the systemic circulation. 

### 3.2. Modeling Dissolution Data

First order and Weibull dissolution models were fitted to the in vitro dissolution data, but the best fit was obtained with the Weibull model. Table 2 summarizes the kinetic parameters and several indexes of goodness of fit. As it can be seen for both formulations, the Weibull model provided a statistically significant better fit.

Correlation coefficients of the Weibull model were higher than the ones obtained with the First order model. AIC values were lower for the more complex model, indicating a better fit than the simple model. The sum of squared residuals was clearly smaller for the Weibull function. When residual variances (Sum of squared corrected by their degrees of freedom) were compared through F test, the comparison indicated a statistically significant improvement with the Weibull model versus the First order equation. As the best model was the same for both formulations, it also indicated a similar dissolution mechanism. 

The dissolution media and method used in this study were selected based on previous reports, indicating its biopredictive ability and previously developed IVIVC. Taking into account the reported values for CBZ solubility (2.96 mg/mL in deionized water containing 1% of SLS [29] and 3.412  ±  0.13 mg/mL in the same media [30]) 900 mL of media does not provide sink conditions for a 400 mg tablet. The maintenance of sink conditions is, in general, required in quality control media, but for IVIVC, non-sink conditions can be of application. In actuality, non-sink conditions can be more reflective of the in vivo environment in which the available volume for dissolution is less than 500 mL [31]. Other authors have recently proposed a non-sink dissolution permeation method to discriminate among CBZ formulations with and IVIVC developed with mice data [32]. 

When in vitro fractions dissolved versus time plots and in vivo fractions absorbed versus time were compared, it is obvious that the time scales are different, i.e., dissolution is completed in less than 4 h while in vivo absorption took almost 20 h to be completed. In consequence, the direct correlation of fractions absorbed versus fractions dissolved is not possible without the time scaling procedure.

### 3.3. In Vitro–In Vivo Data Modeling

To establish the ivivc it is necessary to establish, first of all, a Levy-plot 

Table 3 in columns 1 to 4 summarizes the calculations to obtain the in vitro times that can be used to construct the Levy plot represented in Figure 4.

Table 4 represents the in vitro original times scaled to in vivo times using the Levy plot. Once the in vitro times have been scaled up to in vivo times, then the scaled dissolution profile, represented in Figure 5, can be used to construct the Level A IVIVC.

The scaled profiles were again used to fit the Weibull model and obtain the scaled Weibull parameters which are summarized in Table 5.

With the new Weibull parameters, it is possible to estimate the fraction dissolved at any time. Table 3 in column 5 shows the in vitro dissolved fractions estimated at the same equivalent in vivo times with the new scaled Weibull parameters.

The data in Figure 6 represent the final two-step Level A IVIVC.

The linear correlation depicted in Figure 6 presents a good determination coefficient (R^2^) and it is clearly a single relationship for both formulations. Both aspects indicate in the first place that dissolution is the limiting step for the input of CBZ in the systemic circulation, and—in second place—that the in vitro method, despite its simplicity, reproduced the in vivo dissolution. 

### 3.4. Two-Step IVIVC Predictability

In order to check the predictability of the correlation, the theoretical fractions absorbed were estimated from the fractions dissolved through the linear relationship (see Figure 6). The theoretical or predicted Fabs were back-transformed in plasma concentrations represented in Figure 7 and Table 3 (column 7), and the prediction error of Cmax and AUC was estimated. 

Prediction errors of the two-step approach are summarized in Table 6.

From a regulatory point of view, the prediction errors were within the accepted limits (up to 15% for each formulation with an average of 10% for all formulations) [4,5]. 

In conclusion, the in vitro dissolution profiles can be used to predict plasma levels of CBZ from its IR products.

### 3.5. IVIVC One-Step 

In Figure 8 the experimental and predicted values in vitro and in vivo with the one-step approach are displayed.

The prediction errors with the one-step IVIVC approach are summarized in Table 7.

Parameters of the model are summarized in Table 8.

For the one-step procedure, the prediction errors were slightly lower for all the parameters, and it is interesting to observe that the plasma profiles are better captured in the one-step procedure compared with the two-step one (see predicted plasma profiles in Figure 7 versus Figure 8). Nevertheless, from the standpoint of prediction errors, both approaches are adequate and allowed for the prediction of the plasma levels from the dissolution profiles. In the EMA guidance [5], the recommended procedure is the one-step IVIVC using a mechanistic model. Nevertheless, it is also advised that constructing, as a first approach, the two-step calculations might give some insights into the mechanistic relationship between in vitro and in vivo dissolution such as, for instance, the need for a time-scaling factor detected with the Levy’s plot. FDA guidance [4] requests the two-step procedure, which might not be possible to construct. For instance, for some complex relationships between in vitro and in vivo dissolution/absorption when a preferential dissolution segment or absorption windows exist, a single IVIVC linear two-step correlation might be challenging. In those complex situations, connecting the in vitro dissolution with in vivo plasma profiles requires a mechanistic model defined with differential equations [33,34].

### 3.6. Significance of the Results

CBZ is a non-ionizable molecule whose solubility does not change with pH [35]. Its permeability is high in human intestines compared to metoprolol [36] and passive diffusion is the main absorption mechanism due to its lipophilicity. In accordance with these characteristics, its dissolution process is not affected by the transit to one intestinal segment to the next, and the dissolved amounts are absorbed at a similar rate in the different intestinal segments. In consequence, the challenges for the classical two-step IVIVC approach are minimal as long as the in vitro dissolution method reflects the in vivo dissolution process [37]. In this work, similar predictive performance was achieved with both mathematical approaches. On the other hand, the mechanistic model used in the one-step approach only used a time scaling factor for the in vitro dissolution parameters, which can be directly convoluted with the disposition CBZ parameters without the need of any other kinetic feature.

The differences between formulations in vitro and in vivo may be explained by the difference in particle size distribution. As CBZ is a BCS class II drug, particle size as well as polymorphism are key topics to be investigated in order to ensure the bioequivalence between prototypes and reference products [38]. Both products have the same polymorphic form (form III) which was characterized by X-Ray Diffraction (in-house data). The work performed by Sehić S. et al. [39] demonstrated that commercial samples from the same polymorphic form III (anhydrous) presented particles with different morphology and size distribution. Those differences clearly impacted the kinetics of conversion from anhydrous to the dihydrate CBZ and the dissolution behavior of their formulations.

Moreover, in order to figure out the likely influence of excipients in the in vitro and in vivo behavior of both formulations, a dissolution profile was carried out with crushed tablets of both products in the media containing 1% of SLS (in-house data, not shown). The same difference with intact tablets was observed in the dissolution profiles with crushed tablets, and in consequence, differences in disintegration can be ruled out as a potential source of dissolution differences. Therefore, it is possible to infer that API characteristics may play a significant role in the in vitro and in vivo behavior of the CBZ formulations used in this study. On the other hand, it is not possible to completely rule out the role of excipients, as they might influence, for instance, particle wettability. A previous clinical study, using the same reference product and a test product manufactured with smaller particle size showed also statistically significant Cmax differences in epileptic patients. [40].

The assayed pharmaceutical products used in this work were formulated with common and non-problematic excipients (those affecting transit time or permeability) CBZ dissolution from these products is driven mainly by CBZ particle size. In combination with the previously discussed physicochemical characteristics, a simple unbuffered media (but with SLS at 1% to ease particle wetting and ensure complete dissolution) might reproduce the in vivo dissolution. Even if the in vivo dissolution environment is much more complex, most of the in vivo factors (as pH changes during transit, bile secretions presence) do not affect the in vivo dissolution of CBZ. Consequently, these factors do not need to be incorporated in the in vitro dissolution model, which can be kept as simple as possible.

As this medium was able to distinguish between differences in API particle size, it could be a highly relevant tool in Latin-American countries to ensure and check the quality of their oral CBZ IR marketed products for which BE studies have not been requested by most health authorities [41].

In this work, it was shown that both approaches, two-step, recommended in FDA guidance, and one-step, recommended in EMA guidelines, provided good in vivo predictability for CBZ, a BCS class II drug. 

This article reinforces the evidence that, for CBZ, the same dissolution medium can be used during the product development as well as for quality control purposes. Some authors have suggested approaches for establishing the link between the dissolution test and in vivo performance [42,43]. Therefore, this is a unique situation, as in general, quality control (QC) dissolution methods are of application only to a particular pharmaceutical product to check the consistence of the manufacturing process. QC methods do not have, in most cases, any biopredictive aim, while in the case of CBZ products, it would be possible with the QC method to correlate possible deviations in the manufacturing process (discriminatory power) and their in vivo impact (biopredictive relevance) for a narrow therapeutic index drug, such as CBZ. 

## 4. Conclusions

As proposed by previous studies, the dissolution method in apparatus USP II at 75 rpm with 900 mL of aqueous media containing 1% of SLS was successfully used to develop a Level A IVIVC with two CBZ oral IR products which provides additional evidence that this medium can be classified as a biopredictive dissolution tool for CBZ oral IR products with conventional excipients. On the other hand, we confirmed the similar outcome of one-step versus two-step procedures for IVIVC in an uncomplicated drug (constant solubility, no absorption window, no carrier mediated absorption or saturated metabolic step).

## Figures and Tables

**Figure 1 pharmaceutics-12-00558-f001:**
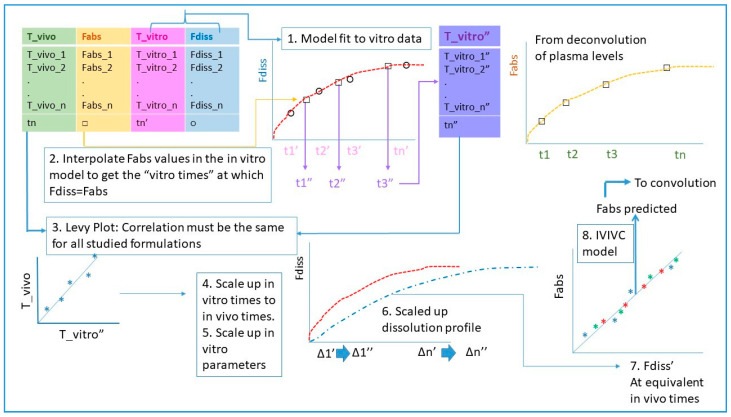
Steps describing the calculations to establish a two-step in vitro–in vivo correlations (IVIVC) when in vivo and in vitro dissolution processes take place at different rate and in consequence the time scale is different.

**Figure 2 pharmaceutics-12-00558-f002:**
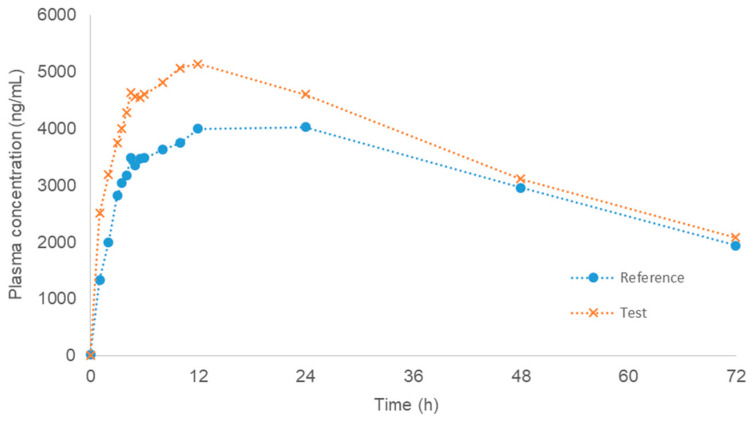
Reference and test average plasma levels (*N* = 11, pilot bioequivalence study).

**Figure 3 pharmaceutics-12-00558-f003:**
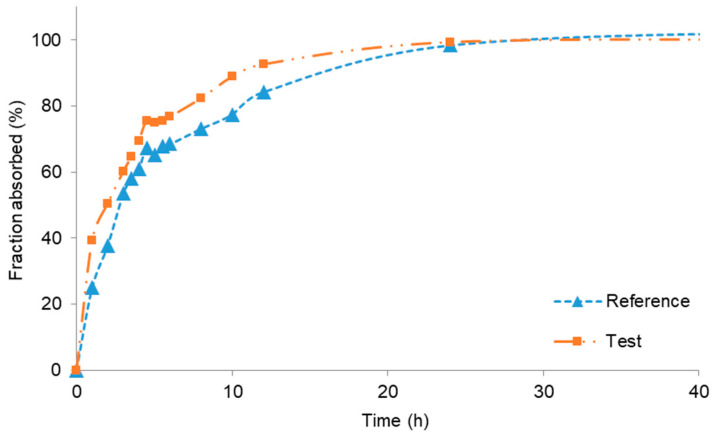
Bioavailable fractions (usually described as Fractions absorbed) obtained by Wagner–Nelson analysis from reference and test formulations.

**Figure 4 pharmaceutics-12-00558-f004:**
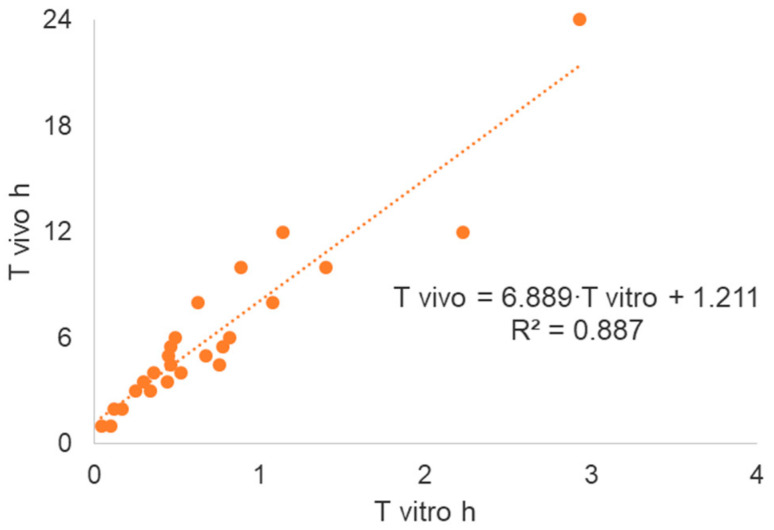
Levy plot constructed with data in Table 3. Times up to 24 h were used for test formulation and up to 12 h for reference formulation. Later points were excluded, as the relationship was no longer univocal.

**Figure 5 pharmaceutics-12-00558-f005:**
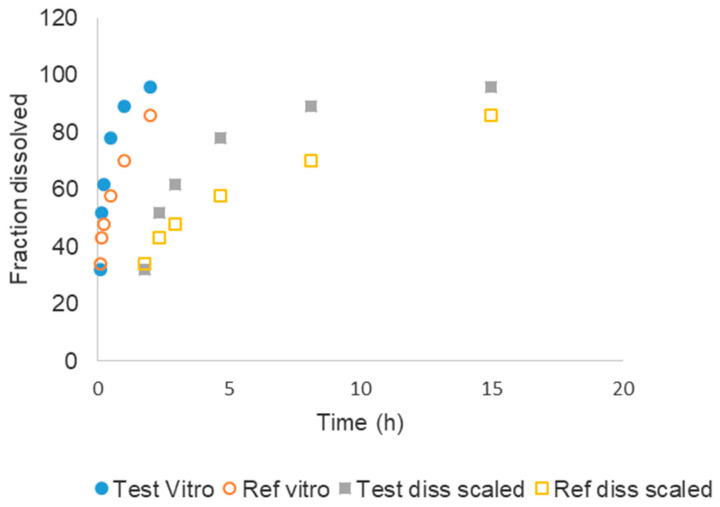
Dissolution profiles in their original in vitro time scale and scaled to equivalent in vivo times (see Table 4) with the Levy plot.

**Figure 6 pharmaceutics-12-00558-f006:**
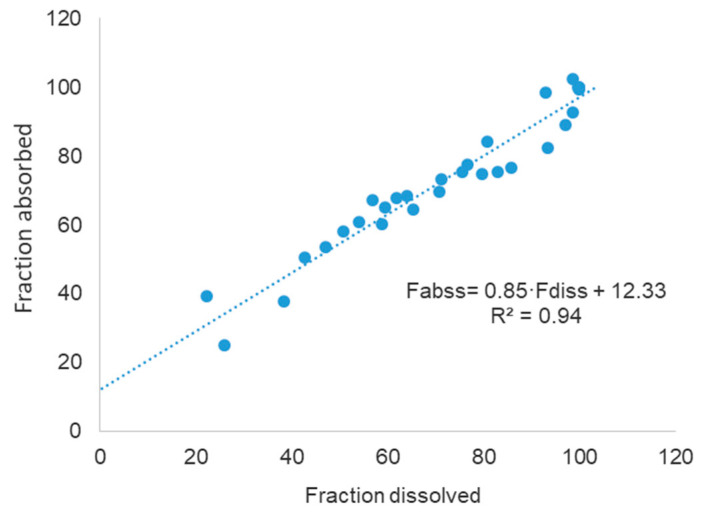
Two-step IVIVC model.

**Figure 7 pharmaceutics-12-00558-f007:**
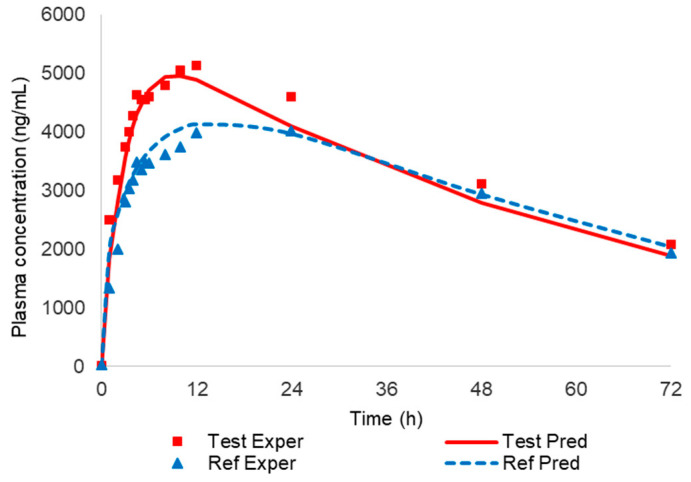
Experimental and predicted plasma levels of both carbamazepine (CBZ) formulations with the two-step IVIVC Level A approach.

**Figure 8 pharmaceutics-12-00558-f008:**
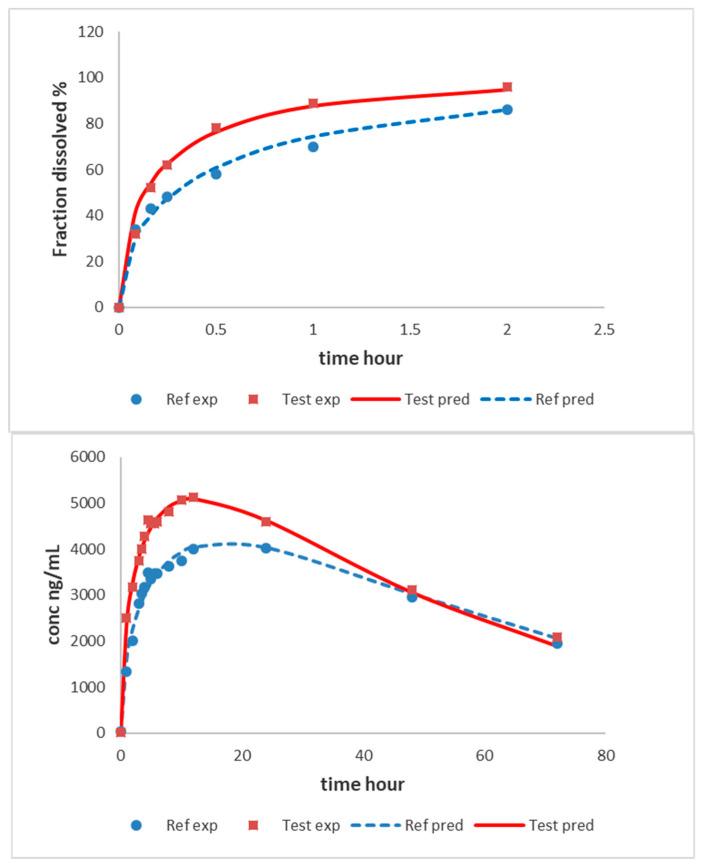
Experimental (symbols) and predicted (lines) in vitro fraction dissolved and plasma levels from reference and test formulations with the one-step IVIVC Level A approach.

**Table 1 pharmaceutics-12-00558-t001:** Summary results of the pilot bioequivalence study (*N* = 11).

Parameter *	Geometric Mean Ratio %	90% CI	Power (1-Beta) %	CV_ws_ %
C_max_	126.54	118.33–135.31	99.9	8.55
AUC_t_	117.42	110.88–124.34	99.9	7.31
AUC_inf_	111.60	104.34–119.37	99.9	8.59

* Parameters logarithmically Ln-transformed. Cmax, maximum plasma concentration; AUCt, area under the concentration-time curve from 0 to 96 h; AUCinf, area under the concentration–time curve extrapolated to infinity; CI, confidence interval; CVws, coefficient of variation within subject. Detailed equations for calculations of parameters (geometric mean ratio, CI and CV_ws_ are included in Appendix A).

**Table 2 pharmaceutics-12-00558-t002:** Fitted parameters of Weibull function and First order model to the in vitro dissolution data. Indexes of goodness of fit: R_obs_pred: correlation coefficient of experimental versus predicted values. AIC: Akaike’s information criteria. SS: sum of squared residuals. Df: degrees of freedom estimated as number of data minus number of parameters. Ftab: tabulated F value for 0.05 probability. Fcal: calculated F value.

Parameter	Weibull		First Order	
	Test	Reference	Test	Reference
α	0.418	0.787	/	/
β	0.687	0.47	/	/
kd(h^−1^)	/	/	3.837	2.122
R_obs-pre	0.996	0.994	0.995	0.982
AIC	22.972	22.246	33.532	43.880
SS	23.618	20.926	191.575	1074.971
df	4	4	5	5
Ftab(0.05;1:4)	7.709			
Fcalc Test	28.445	Fcal > Ftab	Weibull model is the best	
Fcalc Ref	201.482	Fcal > Ftab	Weibull model is the best	

**Table 3 pharmaceutics-12-00558-t003:** Columns 1 to 4: Calculations for the Levy plot. Second and third column are the original in vivo data. Fourth column, “t vitro” are obtained from the Weibull function used to fit the in vitro profiles of each formulation using Equation (4) (see step 2 in Figure 1) and interpolation fraction absorbed values, i.e., in the in vitro dissolution profile it takes 0.1 h to get 39.23% dissolved. Column 5 fractions dissolved estimated at the original in vivo sampling times. Column 3 and 5 were used to represent the IVIVC. Columns 6 and 7 predicted fractions absorbed from the IVIVC linear relationship and the back-calculated predicted plasma concentrations.

1	2	3	4	5	6	7
Formulation	T Vivo (h)	Fabs Vivo	T Vitro Equivalent to T Vivo (h)	Interpolated Fdiss	Fabs Predicted with IVIVC	Conc Predicted (ng/mL)
Test	1	39.23	0.10	22.33	0.31	1829.6
Test	2	50.50	0.17	42.76	0.48	2813.7
Test	3	60.14	0.25	58.80	0.62	3564.5
Test	3.5	64.55	0.30	65.27	0.68	3858.3
Test	4	69.50	0.36	70.83	0.72	4104.3
Test	4.5	75.54	0.46	75.57	0.76	4307.9
Test	5	74.89	0.45	79.60	0.80	4474.5
Test	5.5	75.40	0.46	83.01	0.83	4608.8
Test	6	76.74	0.49	85.89	0.85	4715.3
Test	8	82.36	0.63	93.41	0.91	4938.1
Test	10	88.90	0.88	97.01	0.94	4960.6
Test	12	92.68	1.14	98.68	0.96	4887.2
Test	24	99.34	2.93	99.99	0.97	4096.8
Test				100.00	0.97	2785.3
Test				100.00	0.97	1893.4
Ref	1	24.90	0.04	25.96	0.34	2009.8
Ref	2	37.73	0.12	38.31	0.45	2590.2
Ref	3	53.43	0.34	47.14	0.52	2987.1
Ref	3.5	58.02	0.44	50.76	0.55	3143.6
Ref	4	60.96	0.53	53.99	0.58	3279.4
Ref	4.5	67.24	0.76	56.89	0.60	3397.9
Ref	5	65.22	0.67	59.52	0.63	3501.8
Ref	5.5	67.72	0.78	61.91	0.65	3593.2
Ref	6	68.49	0.82	64.10	0.67	3673.8
Ref	8	73.18	1.08	71.28	0.73	3911.7
Ref	10	77.45	1.40	76.62	0.77	4052.2
Ref	12	84.31	2.22	80.72	0.81	4127.4
Ref *	24	98.46	12.53	92.90	0.91	3964.7
				98.58	0.96	2933.2
				99.63	0.97	2038.4

* Indicated excluded value for the Levy plot.

**Table 4 pharmaceutics-12-00558-t004:** Vitro times scaled to vivo times using the Levy plot.

Formulation	Original t Vitro (hours)	Scaled t Vitro to t Vivo (Hours)
Test	0.083	1.785
Test	0.167	2.359
Test	0.250	2.933
Test	0.500	4.655
Test	1.000	8.099
Test	2.000	14.988
Ref	0.083	1.785
Ref	0.167	2.359
Ref	0.250	2.933
Ref	0.500	4.655
Ref	1.000	8.099
Ref	2.000	14.988

**Table 5 pharmaceutics-12-00558-t005:** Weibull parameters of the in vitro profile scaled up to the in vivo times.

Parameter	Test	Parameter	Reference
α	3.958	α	3.327
β	1.143	β	0.684

**Table 6 pharmaceutics-12-00558-t006:** Predictability analysis of the two-step IVIVC approach.

Parameter	Experimental	Predicted	% Error
Cmax Test (ng/mL)	5135.86	4960.58	3.41
Cmax Ref (ng/mL)	3995.05	4127.41	−3.31
AUC_t_ Test (ng/mL∗h)	388,931.0	355,617.0	8.57
AUC_t_ Ref (ng/mL∗h)	354,986.6	364,858.3	−2.78

**Table 7 pharmaceutics-12-00558-t007:** Predictability analysis of the one-step IVIVC approach.

Parameter	Experimental	Predicted	% Error
Cmax Test (ng/mL)	5135.86	5093.17	0.83
Cmax Ref (ng/mL)	3995.05	4053.67	−1.47
AUC_t_ Test (ng/mL∗h)	388,931.0	374,575.0	3.69
AUC_t_ Ref (ng/mL∗h)	354,986.6	368,207.0	−3.72

**Table 8 pharmaceutics-12-00558-t008:** Parameters of the model.

Parameter	Estimate	Standard Error
alfaT	0.455	0.722
betaT	0.555	1.269
scalfa	5.627	8.969
alfaR	0.717	1.138
betaR	0.542	1.240
scbeta	1.039	2.377
kel h^−1^	0.022	0.003
Vd mL	51,686.649	6498.458

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
