# Peer review of "In Vivo Predictive Dissolution (IPD) for Carbamazepine Formulations: Additional Evidence Regarding a Biopredictive Dissolution Medium"

_pharmaceutics, 2020, doi:10.3390/pharmaceutics12060558_

Round 1

Reviewer 1 Report

The paper describes the development of an In vitro-in vivo correlation of two IR formulations of carbamazepine. 

In my opinion the interest for the readers can be found in the comparison between the two approaches of correlation more than in the evaluation of the medium as biorelevant. I would suggest to modify the title to eliminate the biorelevant value of the medium. Previous literature, correctly cited by the authors,  already discussed this aspect  and also reported that CBZ in vivo behaviour depends more on pharmacokinetic behaviour than from in vitro release.

Moreover, some more discussion is necessary to improve the paper.

Some minor observations:

Tables 2 and 5 are not clearly presented: the parametrs that are compared for test and reference are the same? alpha and beta coefficients?

Table 3 is not clearly presented: it is not clear for the reader the passage from column 3 and column 7, for example.

Comment of Figure 2 says that the plt is restricted to 40 hours (is it?)

Line 160 "in-vivo times" were included up to 2 hours (correct?)

Some mispelling are present and should be corrected. 

Author Response

we would like to agree the reviewer comment because the paper has improved considerably

attach you can see the point by point with answer and the information that we have added in the final paper

Reviewer 2 Report

Overall, the manuscript is well written and of interest to the readers of Pharmaceutics. I think the methodological section is well described and it can serve to other authors to perform IVIVC analyses. However, there are few drawbacks before publication that require revision:

BE -page 4,line 25, which type of ANOVA was performed, one tail or two tails?

-it is considered that CBZ follows a mono compartimental model when actually its half life is pretty high. Please, reconsider the statement made in the text.

The therapeutic index for CBZ is pretty low, should not be the acceptance criteria for BE between 90-111%?

Which pH does the dissolution media have? Would not be important to keep stable the pH of the media using a buffer rather than deionised water?

I think the other models used for fitting the dissolution data should be also mentioned in the text along with Weibull.

the section regarding the Scale factors should be described in more detail.

In Fig. 1, error bars should be added.

In Table 1, please explain in the table caption the meaning of geometric mean ratio, power, CV and CI. Also describe the formula employed to calculate each of these parameters.

I think is important to add the XR data into the main text as maybe slight differences in the degree of crystallinity May have a great impact on dissolution.

In eq 3, please defined well all the terms that appear in the equation as may not be evident for the reader to ensure full understanding.

Finally, I think the discussion should be extended. Currently, it is very minimal and reading some of the references used for the authors (eg. 12) I wonder which is the novelty of the current paper. Please state clear the novelty of this paper and which new piece of information adds to the world knowledge on BE.

Author Response

thanks for the comments. You helped us to improve the paper

Reviewer 3 Report

The manuscript presents information that supports the use of a USP dissolution medium as a universal biopredictive medium. However the data of this study, as well as the references 10 and 11 refer to the case of carbamazepine only. Therefore, there is no sufficient evidence to support the term "universal". Also there is no discussion of the possible reasons that render this particular dissolution medium, suitable for biopredictive testing, and there is no sufficient description in the materials and methods, other than the fact that it contains 1% SLS. Finally, the explanation of the differences between test and reference, based on different particle size of the API (lines 301-302), is merely a speculation that is not supported by evidence.

Author Response

Thanks so much for your suggestions.

The manuscript presents information that supports the use of a USP dissolution medium as a universal biopredictive medium. However the data of this study, as well as the references 10 and 11 refer to the case of carbamazepine only. Therefore, there is no sufficient evidence to support the term "universal".

The adjective universal has been deleted. We understand it may be confusing. We were not claiming this media can be universally used for al drugs, but for all IR carbamazepine products.

Also there is no discussion of the possible reasons that render this particular dissolution medium, suitable for biopredictive testing,

The reasons are that being CBZ a non ionisable drug formulated with common non problematic excipients (those affecting transit time or permeability) CBZ dissolution from these products is driven mainly by CBZ particle size, and a simple unbuffered media (but with SLS at 1% to ease particle wetting) might reproduce the in vivo dissolution. Even if the in vivo dissolution environment is much more complex, most of the in vivo factors (as pH changes during transit, bile secretions presence) do not affect in vivo dissolution of CBZ then these factors do not need to be incorporated in the in vitro dissolution model which can be kept as simple as possible.

and there is no sufficient description in the materials and methods, other than the fact that it contains 1% SLS.

Because the media is as simple as that. Deionised water with 1% SLS which is the quality control media indicated in the USP.

Finally, the explanation of the differences between test and reference, based on different particle size of the API (lines 301-302), is merely a speculation that is not supported by evidence.

We have change this paragraph to the following:

The same difference with intact tablets has been observed in the dissolution profiles with crushed tablets, in consequence differences in disintegration can be ruled out as a potential source of dissolution differences. Therefore, it is possible to infer that API characteristics may play the main role in the in vitro and in vivo behavior of the CBZ formulations used in this study even if it is not possible to completely ruled out the role of excipients as they might influence for instance particle wettability

Reviewer 4 Report

The authors report Level A in-vitro-in vivo correlations for two different carbamazepine formulations using one-step and two-step approaches with 1% SLS aqueous media. The topic may be interesting to the readers of the journal. However, major drawback of the paper is poor writing. Some information is not clear. For example, the authors use 900 mL of 1% SLS aqueous solution. But it is not clear why the authors used this media and whether it can maintain sink conditions. What is the solubility of the drug in the media? In addition to this the following should be considered to improve the paper. As a result, I consider the paper is not suitable for publication in its current form. A considerable modification is required prior to reconsidering the paper for publication.

  1. In all equations please clearly use subscript or superscript. Also use of capital letters should be consistent.
  2. In Figure 1, please indicate standard deviations for each data.
  3. Explanation on Bioequivalence results should be added before Table 1. The results part should explain key findings and discussion needs to be added to analyze the results obtained
  4. What is the blank of Table 2?
  5. Sections 3.1, 3.2, 3.3, 3.4, 3.5 should be re-written as those sections seem to be incomplete. There should be appropriate explanation on the results obtained and proper discussion of the results. The authors simply demonstrate Tables and Figures without explaining the key findings and scientific meaning.

Author Response

Thanks for your comments. your contribution help us to improve our paper considerably

Round 2

Reviewer 1 Report

I think the authors correctly addressed my observations, and the present version is suitable for publication

Reviewer 2 Report

I believe the authors have addressed all comments suggested by the reviewers and hence, the paper is ready for publication in Pharmaceutics. 

Reviewer 3 Report

The authors have provided the necessary clarifications and met the referees' points by suitable alteration of the text, therefore I recommend acceptance for publication.

Reviewer 4 Report

The authors appropriately revised their original version to address all the comments made to them. Thus, I consider the revised paper can now be published without further modification.